# COVID-19 Subunit Vaccine with a Combination of TLR1/2 and TLR3 Agonists Induces Robust and Protective Immunity

**DOI:** 10.3390/vaccines9090957

**Published:** 2021-08-27

**Authors:** Soo-Kyung Jeong, Yoon-Ki Heo, Jei-Hyun Jeong, Su-Jin Ham, Jung-Sun Yum, Byung-Cheol Ahn, Chang-Seon Song, Eun-Young Chun

**Affiliations:** 1R&D Center, CHA Vaccine Institute, Seongnam-si 13493, Korea; soodding@chamc.co.kr (S.-K.J.); hyk0345@chamc.co.kr (Y.-K.H.); gkatn0217@chamc.co.kr (S.-J.H.); jsyum@chamc.co.kr (J.-S.Y.); bcahn@chamc.co.kr (B.-C.A.); 2College of Veterinary Medicine, Konkuk University, Seoul 05029, Korea; nar21ss@hanmail.net

**Keywords:** SARS-CoV-2, subunit vaccine, adjuvant, TLR agonist, neutralizing antibody, IFN-γ-producing T cell, cell-mediated immunity, ferret model, mucosal immunity

## Abstract

The development of COVID-19 vaccines is critical in controlling global health issues under the COVID-19 pandemic. The subunit vaccines are the safest and most widely used vaccine platform and highly effective against a multitude of infectious diseases. An adjuvant is essential for subunit vaccines to enhance the magnitude and durability of immune responses. In this study, we determined whether a combination of toll-like receptor (TLR)1/2 and TLR3 agonists (L-pampo) can be a potent adjuvant for severe acute respiratory syndrome coronavirus 2 (SARS-CoV-2) subunit vaccine. We measured a neutralizing antibody (nAb) and an angiotensin-converting enzyme 2 (ACE2) receptor-blocking antibody against SARS-CoV-2 receptor-binding domain (RBD). We also detected interferon-gamma (IFN-γ) production by using ELISPOT and ELISA assays. By employing a ferret model, we detected nAbs and IFN-γ producing cells and measured viral load in nasal wash after the challenge of SARS-CoV-2. We found that SARS-CoV-2 antigens with L-pampo stimulated robust humoral and cellular immune responses. The efficacy of L-pampo was higher than the other adjuvants. Furthermore, in the ferret model, SARS-CoV-2 antigens with L-pampo elicited nAb response and antigen-specific cellular immune response against SARS-CoV-2, resulting in substantially decreased viral load in their nasal wash. Our study suggests that SARS-CoV-2 antigens formulated with TLR agonists, L-pampo, can be a potent subunit vaccine to promote sufficient protective immunity against SARS-CoV-2.

## 1. Introduction

The new emergence of severe acute respiratory syndrome coronavirus 2 (SARS-CoV-2) has caused a worldwide pandemic since early 2020. The World Health Organization (WHO) reported that 141 million cases of COVID-19 were confirmed and over 3 million died worldwide as of April 2021 [1]. To control the spread of COVID-19 globally, the development of safe and effective vaccines is urgently needed. Burgeoning global efforts have developed several vaccines based on mRNA technology or a virus-vector platform. However, these vaccines are not able to meet global vaccination requirements, and their long-term efficacy and safety concerns are under investigation [2]. Moreover, growing evidences indicate that recovered patients cannot be completely excluded from the population requiring vaccination due to reinfection or SARS-CoV-2 variants [3,4]. Therefore, alternative vaccine approaches against SARS-CoV-2 are constantly required.

Of several vaccine platforms, protein-based subunit vaccines are well-known as the safest and widely used vaccine development platform. They have been highly effective against multiple infectious diseases such as hepatitis-B, diphtheria, pertussis, tetanus, and shingles [5]. In addition, the use of adjuvanted subunit vaccine platforms could benefit special populations such as infants and the elderly with a demonstrable history of safety and efficacy in such populations [6]. However, the immunogenicity of subunit antigens is relatively weak, and it is hard to elicit strong immune response when used alone. Therefore, it is inevitable to consider employing an adjuvant system to boost and sustain immune responses by vaccination. An adjuvant is an immune-stimulatory component which enhances the magnitude and durability of immune responses induced by vaccination even with low doses of antigens [5]. In general, Alum is the most commonly used adjuvant system and strongly induces humoral immune responses. Novel adjuvants, such as emulsion adjuvants (e.g., MF-59 and the AS0 adjuvant systems) or toll-like receptors (TLRs) agonists, have been developed for optimal and efficient vaccine formulation against influenza, hepatitis B virus (HBV), human papillomavirus (HPV), or varicella zoster [7,8,9]. The most recent study has demonstrated that these adjuvants promote protective immunity against SARS-CoV-2 in non-human primates [10].

As a potent adjuvant system, we have developed L-pampo, a proprietary adjuvant composed of TLR1/2 and TLR3 agonists [11]. Our prior study reported that L-pampo induces stronger humoral immune response against HBV than Alum, and L-pampo also affects cell-mediated immune responses such as increased multifunctional CD4^+^ T cells [11]. Therefore, we have hypothesized that L-pampo can be a potent adjuvant to induce strong immune responses against SARS-CoV-2.

In this study, we showed that SARS-CoV-2 antigens with L-pampo elicited robust humoral and cellular immune responses against SARS-CoV-2. Receptor-binding domain (RBD) and S1 antigens or RBD-Fc with L-pampo induced higher neutralizing antibody (nAb) and angiotensin-converting enzyme 2 (ACE2) receptor-blocking antibody production compared to widely used adjuvants (Alum, AddaS03, and AddaVax). In addition, L-pampo highly increased cell-mediated immune responses such as RBD-specific IFN-γ-secreting cells. Furthermore, by using the ferret model which is a suitable animal model for SARS-CoV-2 infection, we found that RBD and S1 with L-pampo elicited strong nAb response against SARS-CoV-2 and increased both RBD- and S1-specific cellular immune responses. After virus challenge, ferrets immunized with L-pampo and RBD and S1 antigens rapidly decreased viral load in their nasal wash. Our study suggests that L-pampo, a combination of TLR1/2 and TLR3 agonists, can be a potent adjuvant system for subunit vaccines against COVID-19.

## 2. Materials and Methods

### 2.1. Animal Studies

#### 2.1.1. Mice

Female BALB/c mice aged 6 weeks were purchased from the Orient Bio Inc. (Seongnam, Korea) and housed in a specific-pathogen-free animal facility at CHA University (Seongnam, Korea). All mice were handled in accordance with the standards approved by the Institutional Animal Care and Use Committee (IACUC; #200081) of CHA University.

#### 2.1.2. Ferrets

Sixteen female ferrets aged 12 months were purchased from ID Bio Corporation (Cheongju, Korea). The ferrets were tested seronegative for SARS-CoV-2 by serum neutralization test and housed in animal biosafety level 3 (ABSL3) facilities within the Konkuk University (Seoul, Korea). All ferret care was performed strictly according to the animal care guidelines and experiment protocols approved by the Institutional Animal Care and Use Committee (IACUC) of Konkuk University (permit number: KU20074-1).

### 2.2. Adjuvant Formulation

Recombinant SARS-CoV-2 spike protein (S1) and RBD antigens were generously provided by Bionote (Hwaseong, Korea). Recombinant SARS-CoV-2 RBD-Fc, which contained RBD in S1 of SARS-CoV-2 and Fc fragment of human IgG, was kindly provided by PanGen Biotech Inc. (Suwon, Korea). Alum (vac-alu-250), AddaVax^TM^ (vac-adx-10), and AddaS03^TM^ (vac-as03-10) were purchased from Invivogen (San Diego, CA, USA). CpG1018 (NBP2-31142) was purchased from Novus biologicals (Centennial, CO, USA). For each dose of Alum, an RBD (5 μg) and S1 (5 μg) mixture or RBD-Fc (5 μg) was diluted in 150 mM NaCl and 10 mM sodium phosphate buffer and mixed with Alum (100 μg). AddaVax is a squalene-based oil-in water adjuvant. For each dose, the diluted RBD (5 μg) and S1 (5 μg) mixture or RBD-Fc (5 μg) was mixed with an equal volume of AddaVax (total 100 μL/dose). AddaS03 is a squalene oil-in-water emulsion that is similar to AddaVax but also contains a-tocopherol as an additional component. For each dose, the diluted RBD (5 μg) and S1 (5 μg) mixture or RBD-Fc (5 μg) was mixed with an equal volume of AddaS03 (total 50 μL/dose). CpG1018, a TLR9 agonist, is a synthetic CpG-B class oligonucleotide. For each dose of CpG–Alum, the diluted RBD-Fc (5 μg) was mixed with Alum (50 μg) and incubated on ice for 30 min. After 30 min, 10 μg of CpG were added and mixed rapidly. L-pampo is a mixture of lipopeptide and double-stranded RNA. According to our standard operation procedure (SOP), the diluted RBD (5 μg) and S1 (5 μg) mixture or RBD-Fc (5 μg) was mixed with L-pampo (Low, L) or 3-fold higher doses of L-pampo (High, H). 

### 2.3. Immunization and Virus Challenge

#### 2.3.1. Mouse Immunization

Female BALB/c mice were randomly assigned to each group and immunized intramuscularly with SARS-CoV-2 antigens with or without adjuvants. The total injection volume of each dose was 100 μL, except the case of AddaS03 (50 μL). The mice were boosted with the same antigens and adjuvants at 2 or 3 weeks. The mice were euthanized by the exposure to carbon dioxide (CO_2_) overdose, followed by cervical dislocation as a secondary method of euthanasia. Sera were collected at 2 weeks after boost immunization to detect SARS-CoV-2 Spike RBD-specific antibodies and nAbs as described below. Splenocytes were obtained at 2 weeks after boost immunization to analyze cellular immune responses.

#### 2.3.2. Ferret Immunization and Virus Challenge

For ferret immunization, RBD and S1 (30 μg per each) were admixed with or without L-pampo (total volume: 500 μL) and administered via the intramuscular route in left caudal thigh twice at 2-week intervals. Blood were collected every week for 5 weeks after first immunization. Subsequently, ferrets were intranasally infected with 10^5.5^ 50% tissue culture infective dose (TCID_50_/_mL_) of SARS-CoV-2 (Korea Centers for Disease Control and Prevention; resource no. 43326) under anesthesia. General anesthesia was induced by injection with 7 mg/kg body weight Zoletil (Virbac, Carros, France) and 3 mg/kg body-weight Xylazine-HCl (Rompun; Bayer HealthCare, Leverkusen, Germany) during the sample collection and inoculation of the virus intranasally. Ferrets in the control group were given PBS. Body weight and temperature were measured, and veterinary clinical symptoms were observed every day. After virus challenge, nasal wash was collected on days 2, 4, 6, 8, 10, and 12, according to the procedure used in our previous publication [12]. Ferrets were euthanized by the administration of T-61 (0.5 mL/kg; intravenous route) under general anesthesia, when the experiment was completed.

### 2.4. Neutralizing Assay

The sera from different groups of mice and ferrets were separated and inactivated at 56 °C for 30 min. After two-fold dilution with a complete medium, they were mixed with 200 TCID_50_ of SARS-CoV-2 at a volume ratio 1:1 and incubated at 37 °C for 1 h to neutralize the infectious virus. The mixtures were then transferred to Vero E6 cell monolayers and further incubated at 37 °C for 4 days. The cytopathic effect (CPE) was confirmed with a microscope, and the endpoint-neutralizing titer was evaluated by the dilution number when the CPE became visible at 4 day post-infection.

### 2.5. ACE-2-Blocking Assay

The sera were serially diluted and added at room temperature for 2 h to 96-well ELISA plates pre-coated overnight at 4 °C with 1 μg/mL SARS-CoV-2 RBD protein (Genscript, Piscataway, NJ, USA). After washing plates, HRP-conjugated human ACE-Fc was added to wells and incubated for 1 h at room temperature. After washing, a tetramethylbenzidine (TMB) substrate was added for 10 min, and the plates were read at 450 nm. The percentage of inhibition was calculated from optical density (OD) as follows: (1-OD sample value/OD RBD protein) × 100. Fifty percent of the ACE2-blocking antibody titer was determined by serum dilution, resulting in a 50% inhibition using Sigmaplot. 

### 2.6. SARS-CoV-2-Specific IgG ELISA

Ninety-six-well ELISA microplates were coated with RBD (1 μg/mL) at 37 °C for 2 h and blocked with bovine serum albumin (1%, *w/v*) at 37 °C for 1 h. After the microplates were washed, the diluted sera were added to the plates and incubated at 37 °C for 2 h. Then, the bounded antibodies were detected by incubation with HRP-conjugated anti-mouse IgG(KPL), IgG1, IgG2a, and IgG2b antibodies (Thermo Fisher, Waltham, MA, USA) at 37 °C for 1 h. Reaction was visualized by the addition of a TMB–peroxidase solution (Seracare, Milford, MA, USA). The OD at 450 nm was measured with an automatic ELISA plate reader (Thermo Fisher, Waltham, MA, USA). 

### 2.7. Enzyme-Linked Immunospot (ELISPOT) Assay

Mouse IFN-γ ELISPOT assay (3321-1HP-2, Mabtech, Stockholm, Sweden) and Ferret IFN-γ ELISPOT assay (3312-4HPW-2, Mabtech, Stockholm, Sweden) were performed according to manufacturer’s protocol. Briefly, immunized mouse splenocytes (5 × 10^5^ cells /well) or ferret peripheral blood mononuclear cells (PBMCs; 3.5 × 10^5^ cells /well) were added to a 96-well ELISPOT plate pre-coated with anti-mouse monoclonal IFN-γ antibodies. Mouse splenocytes were incubated with the PepMix^TM^ SARS-CoV-2(S-RBD) peptide pool (JPT Peptide Technology, Berlin, Germany) (4 μg/mL) at 37 °C for 24 h. Ferret PBMCs were incubated with the RBD (5 μg) or S1 (5 μg) antigen at 37 °C for 24 h. Concanavalin A (Sigma, St. Louis, MO, USA) and the medium alone were included as positive and negative controls, respectively. After stimulation, the wells were washed and incubated with the biotinylated anti-mouse IFN-γ antibody or anti-ferret IFN-γ antibody at room temperature (RT) for 2 h. Subsequently, the plate was incubated with streptavidin-horseradish peroxidase for 1 h, and spot-forming cells (SFCs) were developed with the TMB substrate. The spots were counted using a AELVIS ELISPOT Reader. To quantify positive peptide-specific responses, mean spot counts for negative controls were subtracted from peptide-stimulated samples. The results were presented as SFCs/5 × 10^5^ and SFCs/1 × 10^6^ for mouse and ferret model, respectively. 

### 2.8. ELISA Assay

The splenocytes (1.5 × 10^6^ cells/well) from immunized mice were stimulated with the PepMix^TM^ SARS-CoV-2(S-RBD) peptide pool (JPT Peptide Technology, Berlin, Germany) (4 μg/mL) at 37 °C for 48 h. Non-stimulated splenocytes were used as a negative control. After stimulation, IFN-γ production was measured in the splenocyte supernatants by mouse IFN-γ ELISA kit II according to the manufacturer’s protocol (BD Biosciences, San Jose, CA, USA).

### 2.9. Intracellular Cytokine Staining Assay

The mouse splenocytes were stimulated at 37 °C for 20 h with or without the PepMix^TM^ SARS-CoV-2(S-RBD) peptide pool (JPT Peptide Technology, Berlin, Germany) (2 µg/mL) in the presence of 10 μg/mL brefeldin A (BD Biosciences, San Jose, CA, USA). Cells were stained with a yellow LIVEDEAD fixable dead cell stain kit (Invitrogen, Waltham, MA, USA) and anti-CD3 (Clone 500A2, BD), anti-CD4 (Clone RM4-5, BD), and anti-CD8 (Clone 53-6.7, BD) antibodies. The cells were subsequently fixed and permeabilized using the Cytofix/Cytoperm kit (BD Biosciences, San Jose, CA, USA) and stained with anti-TNF-α (clone MP6-XT22, BD, 1:100) and anti-IL-2 (JES6-5H4, BD, 1:100) antibodies. The cells were acquired on a CytoFLEX (Beckman Coulter, Brea, CA, USA) and analyzed with Flowjo software (Tree Star, Ashland, OR, USA).

### 2.10. Viral Load Analysis 

#### 2.10.1. Infectious Virus Titer (TCID_50_)

Virus titers in nasal washes were detected by using Vero E6 cells. All samples were diluted 10-fold with DMEM media supplemented with an antibiotic-antimycotic solution. The diluted samples were inoculated into Vero E6 cells and incubated at 37 °C for 4–5 days. The CPE was monitored with a microscope every day, and the value of TCID_50_/_mL_ was determined using a Reed and Muench method [13].

#### 2.10.2. qRT-PCR

Total RNA was extracted using a MagNA Pure 96 External lysis buffer (Roche, Switzerland), and qRT-PCR reactions were performed using STANDARD M nCoV Real-Time Detection kit (M-NCOV-01, SD Biosensor, KOREA), which targeted regions of *envelope* (*E*) and *RNA-dependent RNA polymerase* (*RdRp*) [14] according to the manufacturer’s protocol. Briefly, PCR were run at an Applied Biosystems 7500 Real-Time PCR instrument system in a volume of 31 μL containing a 10 μL sample, a 14 μL 2019-nCoV reaction solution, 6 μL RTase mix, 0.5 μL ROX, and 0.5 μL internal control. The PCR conditions were as following: 15 min at 50 °C for reverse transcription, 3 min at 95 °C for initial denaturation, 5 cycles of 5 s at 95 °C and 40 s at 60 °C for pre-amplification, and 40 cycles of 5 s at 95 °C and 40 s at 60 °C for amplification. Viral RNA was shown as a cycle threshold (Ct) value that is inversely proportional to the original relative expression level of the target gene. 

### 2.11. Statistical Analysis

Data were analyzed with Graph Prism 9 software. Data are shown as median ± Interquartile range (IQR) or mean ± SD as noted. For comparisons between more than two groups, one-way ANOVA followed by Tukey’s test was performed. For comparison between two independent groups, a two-tailed Mann–Whitney test was used. No samples were excluded from any experiments performed in this study. Differences of *p* < 0.05 were considered statistically significant. 

## 3. Results

### 3.1. RBD and S1 Admixed with L-Pampo Induces Robust Humoral Responses and Cellular Immune Responses

Humoral immune response, specifically nAb, is important to protect from virus infection. To access the immunogenicity and efficacy of L-pampo against SARS-CoV-2 antigens, we immunized BALB/c mice with RBD (5 μg/mouse) and S1 (5 μg/mouse) antigens formulated with different adjuvants including Alum, AddaVax, AddaSO3, or low (L) and high (H) doses of L-pampo. All immunization was administered via the intramuscular route on day 0 and day 14. The antigen-only group was set as a control (Figure 1A). We detected the nAb from the serum of each mouse on day 28 and found that high doses of L-pampo (H) group elicited higher levels of nAbs compared to the Alum, AddaS03, and antigen-only groups (Figure 1B). We then evaluated RBD- and S1-specific Abs that can inhibit the engagement of SARS-CoV-2 antigens to ACE2, which serves as an entry receptor [15,16,17]. The mice immunized with SARS-CoV-2 antigens and L-pampo (H) showed increased Ab titers that blocked SARS-CoV-2 RBD binding to ACE2, compared to mice immunized with SARS-CoV-2 antigens and formulated with the Alum, AddaVax, or antigen-only groups (Figure 1C). The AddaS03 group was likely to decrease the ACE2-blocking Ab titer but not significant compared to the L-pampo (H) group (Figure 1C). We also evaluated total IgG responses against SARS-CoV-2 antigen, specially RBD protein. The AddaVax, AddaS03, L-pampo (L), and L-pampo (H) groups increased total anti-RBD IgG antibodies, and the L-pampo (H) group was significantly higher than the Alum and antigen-only groups (Figure 1D and Appendix A). T helper 1 cell (Th1) and T helper 2 cells (Th2) immune responses are closely associated with vaccine-induced immunity [18]. Therefore, we compared levels of IgG1 and IgG2a/IgG2b, which are representatives of Th2 and Th1 responses, respectively. The AddaVax, AddaS03, L-pampo (L), and L-pampo (H) groups elicited high levels of IgG1 antibodies, and both the L-pampo (H) and (L) groups significantly produced higher levels of anti-RBD IgG1 compared to the Alum and antigen-only groups. Interestingly, Th1 responses with high IgG2a and IgG2b levels significantly increased in both the L-pampo (H) and L-pampo (L) groups compared to in the AddaVax, AddaS03, Alum, and antigen-only groups (Figure 1D and Appendix A).

Recent studies have reported that humoral immune responses decline in recovered patients over time, whereas T cell responses are maintained after infection and during recovery [19,20], indicating cell-mediated immune response plays a critical role in SARS-CoV-2 vaccination. Given that L-pampo induced higher Th1 responses than Alum, AddaVax, and AddaS03 (Figure 1D), we sought to determine whether L-pampo increases cellular immune responses such as IFN-γ, a principle Th1 cytokine production by using ELISPOT and ELISA assays. As expected, we observed that both the L-pampo (H) and L-pampo (L) groups had higher RBD-specific IFN-γ-secreting cell numbers compared to the Alum, AddaVax, AddaS03, and antigen-only groups (Figure 2A and Appendix A). Consistent with ELISPOT assay data, both the L-pampo (H) and L-pampo (L) groups produced higher levels of IFN-γ than the Alum, AddaVax, AddaS03, and Ag only groups (Figure 2B and Appendix A). These data demonstrate that RBD and S1 admixed with L-pampo induce strong antibody production and cellular immunity against SARS-CoV-2 antigens.

### 3.2. RBD-Fc with L-Pampo Induces Both Humoral and Cellular Immune Responses

To confirm the efficacy of L-pampo for various types of SARS-CoV-2 subunit vaccines, we used RBD-Fc as an antigen and utilized CpG-Alum which is a promising adjuvant for COVID-19 vaccines, and L-pampo (L) to induce sufficient humoral and cellular immune responses, shown in Figure 1 and Figure 2. BALB/c mice were intramuscularly immunized with RBD-Fc with or without adjuvants on day 0 and day 21, humoral and cellular immune responses on day 35 were analyzed (Figure 3A). We first detected nAb titers. The CpG-Alum, AddaVax, and L-pampo (L) groups elicited strong nAbs, while the AddaS03 and Alum groups induced nAb production to a lesser extent (Figure 3B). As expected, the L-pampo (L) group manifested higher ACE2-bloking Ab production that inhibited the binding of RBDs to ACE2 receptors compared to the AddaS03, Alum, and antigen-only groups (Figure 3C). Consistent with the immunization with RBD and S1 admixed with L-pampo, the RBD-Fc with L-pampo (L) group increased total anti-RBD IgG and IgG1 antibody responses compared to the Alum and antigen-only groups (Figure 3D and Appendix A). However, L-pampo (L) group significantly induced IgG2a and IgG2b responses compared to the Alum, AddaVax, AddaS03 and antigen-only groups (Figure 3D and Appendix A). We observed CpG–Alum groups induced humoral immune responses as similar to the L-pampo (L) group (Figure 3B–D). The RBD-Fc with L-pampo (L) group also increased the RBD-specific IFN-γ-secreting cell number compared to the Alum, AddaVax, AddaS03, and antigen-only groups (Figure 4A and Appendix A). The total IFN-γ production in the L-pampo (L) group was higher than in all adjuvant groups, except the case of the CpG–Alum group (Figure 4B and Appendix A). To investigate whether L-pampo induces other T cell-specific cytokine production, we accessed TNF-α or Interleukin (IL)-2 expressing CD4^+^ or CD8^+^ T cells by using intracellular cytokine analysis. We observed that the L-pampo (L) group was likely to increase CD4^+^ TNF-α- and CD8^+^ TNF-α-expressing cells compared to all the other adjuvant groups and the control group (Appendix A). IL-2-expressing T cell numbers did not change between the groups (Appendix A). All these data confirmed that L-pampo consistently induces sufficient humoral and cellular immune responses against SARS-CoV-2, even with low doses and different SARS-CoV-2 antigens. 

### 3.3. RBD and S1 Admixed with L-Pampo Induce Protective Immunity in the Ferret Model 

To determine whether the SARS-CoV-2 antigen formulated with L-pampo induces protective immunity against SARS-CoV-2, we employed a ferret model, which is an effective animal model for SARS-CoV-2 infection [21]. First, we immunized ferrets with RBD (30 μg/ferret) and S1 antigens (30 μg/ferret) admixed with L-pampo (total 500 μL/dose) via the intramuscular route twice on day 0 and day 14. Ferrets with PBS or an antigen only formed a control group. Blood was collected from each ferret on day 0, day 14, day 21, day 28, and day 35 (Figure 5A). Ferrets immunized with RBD and S1 admixed with L-pampo produced strong nAbs from day 21 and prolonged the similar levels of nAb by day 35, given the time period in our experiment (Figure 5B). In contrast, the antigen-only group was able to induce substantial nAbs on day 21 but gradually declined the nAb level by day 35 (Figure 5B). We also analyzed cell-mediated immune responses by using ELISPOT assay on day 35. Ferrets immunized with antigens (RBD and S1) and L-pampo increased both RBD- and S1-specific IFN-γ producing cells from their PBMCs compared to the antigen-only and PBS groups (Figure 5C). 

The immunized ferrets were challenged with 10^5.5^ TCID_50_/mL of SARS-CoV-2 (Korea Centers for Disease Control and Prevention; resource no. 43326) via the intranasal route on day 35 and monitored for clinical symptoms such as body weight, temperature, and mortality (Figure 5A). We collected nasal wash samples from immunized ferrets on day 2, 4, 6, 8, 10, and 12 post-challenge (dpc) and analyzed infectious viral load and viral RNA by using TCID_50_ and qRT-PCR, respectively. Ferrets immunized with antigen and L-pampo rapidly decreased virus titers (TCID_50_) in their nasal wash from 2 dpc, while ferrets immunized with antigen only decreased viral shedding at 6 dpc (Figure 5D). Unimmunized ferrets continued to display infectious viral shedding by 6 dpc. By 12 dpc, infectious viral loads were undetectable in nasal wash from all ferrets (Figure 5D). Viral RNA in nasal wash generally reflected infectious viral loads. Consistent with viral loads data, all ferrets immunized with antigen and L-pampo showed lower viral RNA expression (high CT value represents low expression level) at 2 dpc compared to the antigen-only and PBS groups (31.2 ± 1.9 vs. 21.1 ± 2.2 and 16.8 ± 7.5), continued to decrease viral RNA levels by 6 dpc and led to undetectable viral RNA levels at 8 dpc (Table 1). In contrast, viral RNA was highly detectable in ferrets immunized with antigen only at 2 dpc, sustained by 10 dpc and completely cleared at 12 dpc (Table 1). The PBS group maintained high levels of viral RNA by 12 dpc. There were no significant signs of clinical symptoms (body weight and temperature changes) in any ferrets with or without immunization (Figure 6). These data demonstrated that L-pampo induces both humoral and cellular immune responses in the ferret model and can rapidly protect against SARS-CoV-2 through bolstered immunity by immunization. 

## 4. Discussion

In this study, we demonstrated that L-pampo, a combination of TLR1/2 and TLR3 agonists, elicited robust humoral and cell-mediated immune responses against SARS-CoV-2 antigens. We observed that the immunogenicity of L-pampo was higher than other adjuvants that are known as efficient in various vaccine platforms. Furthermore, L-pampo induced strong and sustainable neutralizing Ab production and antigen-specific cellular immune response and subsequently manifested protective effect in ferrets infected with SARS-CoV-2 virus. 

TLRs are a category of pattern-recognizing receptors that are critical for pathogen recognition. TLR agonists or ligands rapidly activate the innate immune system and, consequently, affect adaptive immunity by regulating dendritic cell (DC) activation and the production of costimulatory molecules and pro-inflammatory cytokines. Therefore, TLR agonists have been extensively studied as vaccine adjuvants for various infectious diseases including SARS-CoV and SARS-CoV-2. The previous studies have reported that CpG, PolyI:C, glucopyranosyl lipid A (GLA), and resiquimod (R837) that are for TLR9, TLR3, TLR4, and TLR7/8 agonists respectively have been evaluated in candidate vaccines against SARS-CoV and these TLR agonists augment antibody production and CD8^+^ T cell responses [22,23]. Hence, under the COVID-19 pandemic, TLR agonists, specially CpG and TLR9 agonists, have been evaluated in SARS-CoV-2 vaccine development and clinical trials. For instance, stable fusion SARS-CoV-2 spike (S-2P) adjuvanted with CpG 1018 and Alum (CpG–Alum) induces potent nAb and minimized Th2-based responses and protects hamsters from SARS-CoV-2 challenge [24,25]. The trimeric form of S protein combined with CpG-Alum also induces high levels of neutralizing nAbs and Th1-based cellular immune responses in nonhuman primates [26]. As expected, we have observed the significant effect of CpG–Alum with high levels of nAbs, Th1-biased immune responses and T-cell-specific IFN-γ production in our mouse model. However, it is more intriguing that the capability of L-pampo in the induction of both humoral and cellular immune responses is equivalent to that of CpG–Alum. Given that L-pampo-formulated vaccine induced mainly IgG2a and IgG2b while the vaccine with Alum induced predominantly IgG1 against the hepatitis B virus surface antigen (HBsAg) [27], we contemplated the SARS-CoV-2 antigen with L-pampo would surpass Alum regarding nAb production and cell-mediated immune response. Indeed, L-pampo is superior to Alum and additional AddaVax and AddaS03 in our study. We may attribute the efficacy of L-pampo to the beneficial effects of the combination of TLR3 and TLR1/2 agonists. A synthetic double-stranded RNA (dsRNA), dominantly produces type I interferon (IFN) and strongly polarizes Th1 immunity [28,29], while a synthetic bacterial lipoprotein, produces pro-inflammatory cytokines such as IL-6 and polarizes Th2 immunity [30,31]. Therefore, L-pampo can induce sufficient and balanced Th1/Th2 responses by properly regulating type I interferon (IFN) and pro-inflammatory cytokine production in the context of types of antigens, host immunity, and administration routes. Harnessing the mechanism of the action of L-pampo will be further investigated, and this will open a new pavement to combine L-pampo with different TLR agonists or other types of adjuvants for the development of the future adjuvant system. 

Besides TLR agonists, oil-in-water emulsion adjuvants such as AddaVax and AS03 (AddaS03 in our study) are promising adjuvants for COVID-19 subunit vaccines. AddaVax and AS03 can elicit both humoral and cellular responses through the activation of myeloid cells [8,32]. Particularly, AS03 is being evaluated in clinical trials as a vaccine adjuvant for different recombinant spike protein COVID-19 vaccines [33]. A recent study also shows that vaccinations with SARS-CoV-2 antigens and AS03 induce rapid amnestic nAbs and protect against SARS-CoV-2 in non-human primates [34]. In addition, another study suggests that RBD–nanoparticle (NP) and AS03 can promote Th1 CD4^+^ T cell response in the non-human primate model [10]. In our mouse model, AS03 indeed fosters humoral response and Th2-biased response but does not induce Th1-mediated cellular immune responses compared to L-pampo. This maybe attributes to the composition discrepancy between AS03 (GSK, Philadelphia, PA, USA) and AddaS03 (Invivogen, San Diego, CA, USA) with the utilization of different animal models or antigen components used for subunit vaccines. 

Ferret has been used as a highly valuable and effective animal model for testing the pathogenicity and transmission of human respiratory viruses, including influenza virus and respiratory syncytial virus [35]. Not surprisingly, the ferret model has been investigated for studies of the COVID-19 pathogenesis and SARS-CoV-2 transmission. Due to predominant upper-respiratory-tract infection in these animals, the ferret model is known as a well-suited model for testing the efficacy of mucosal vaccines and therapeutic agents against SARS-CoV-2 infection [36]. We have observed that L-pampo produces sufficient nAbs after boosting immunization and, more importantly, sustains high levels of the nAbs for at least several weeks. This strong and prolonged antibody production contributes to rapid viral clearance in nasal wash after virus challenge and affords to protection against SARS-CoV-2 infection. In addition to antibody response, cellular immune response also plays a critical role in decreasing the viral load in nasal wash of ferrets. Whether L-pampo modulates virus-specific T cells, memory B cells or innate immune cells in the mucosal environment such as upper and lower respiratory tract and the lung or whether different administration routes of L-pampo such as intranasal affect the L-pampo efficacy to protect against SARS-CoV-2 merits further investigation.

## 5. Conclusions

Collectively, our study provides evidence that L-pampo can be as a substantial and effective adjuvants system that can support the development of an ideal vaccine candidate in mitigating the burden of the global COVID-19 pandemic. In addition, our animal model data support that L-pampo can advance clinical studies to further demonstrate safety, immunogenicity, and efficacy with a variety of SARS-CoV-2 antigens. 

## Figures and Tables

**Figure 1 vaccines-09-00957-f001:**
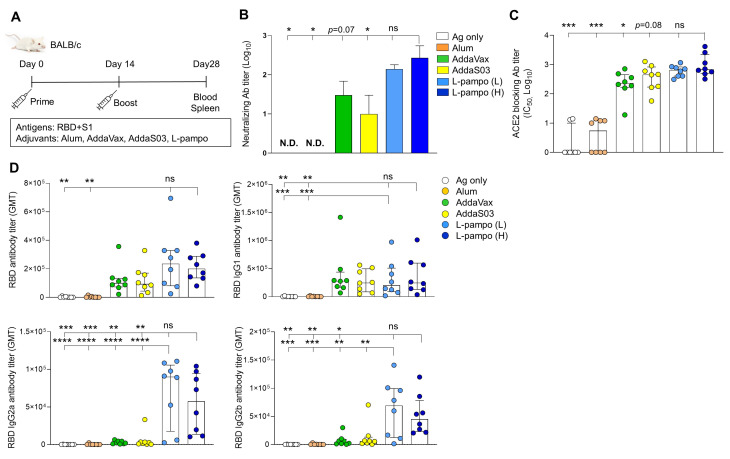
Receptor-binding domain (RBD) and S1 antigens with L-pampo induce robust humoral responses. (**A**) A schematic of immunization strategy. BALB/c mice (total n = 48; n = 8/group) were immunized with severe acute respiratory syndrome coronavirus 2 (SARS-CoV-2) antigens (RBD and S1) with or without adjuvants intramuscularly (i.m.) on day 0 and day 14. (**B**) Neutralizing antibody production on day 28. (**C**) Angiotensin-converting enzyme 2 (ACE2)-blocking antibody titers on day 28. (**D**) RBD-specific total IgG, IgG1, IgG2a, and IgG2b antibodies on day 28. Data shown are median ± Interquartile range (IQR). Each dot represents an individual mouse. GMT in antibody assay indicates geometric mean titers. Data reflect 2 independent experiments (n = 4/group in single experiment). Asterisks indicate statistically significant differences in comparison to the L-pampo (L) or L-pampo (H). * *p* < 0.05; ** *p* < 0.01; *** *p* < 0.001; **** *p* < 0.0001; ns, non-significant, one way ANOVA with the Tukey’s test.

**Figure 2 vaccines-09-00957-f002:**
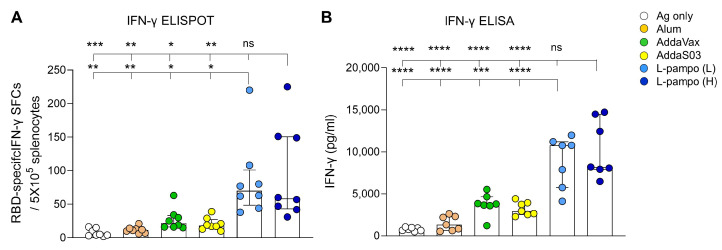
RBD and S1 antigens with L-pampo induce strong cell-mediated responses. (**A**,**B**) BALB/c mice (total n = 48, n = 8/group in (**A**); total n = 42, n = 7/group in (**B**)) were immunized with SARS-CoV-2 antigens (RBD and S1) with or without adjuvants i.m. on day 0 and day 14 as described in Figure 1A. On day 28, splenocytes were stimulated with the PepMix^TM^ SARS-CoV-2(S-RBD) peptide pool, and RBD-specific IFN-γ secreting cells were analyzed as spot-forming cells (SFCs) by ELISPOT assay (**A**) or total IFN-γ production by ELISA assay (**B**). Data shown are median ± IQR. Each dot represents an individual mouse. Data reflect 2 independent experiments (n = 4/group in single experiment). Asterisks indicate statistically significant differences in comparison to the L-pampo (L) or L-pampo (H) groups. * *p* < 0.05; ** *p* < 0.01; *** *p* < 0.001; **** *p* < 0.0001; ns, non-significant, one way ANOVA with Tukey’s test.

**Figure 3 vaccines-09-00957-f003:**
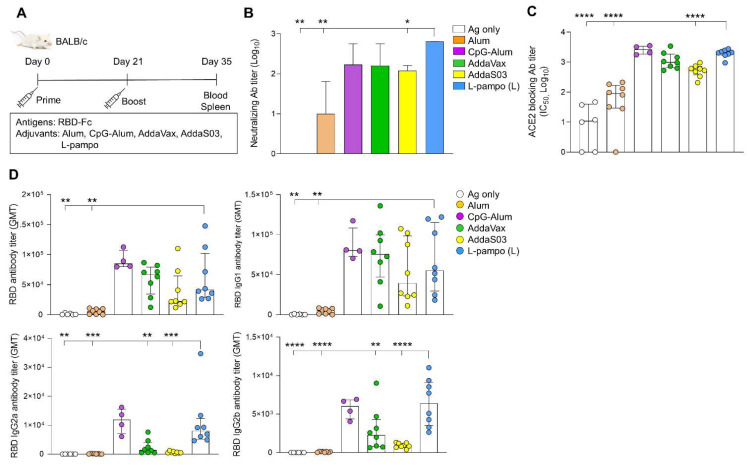
RBD-Fc antigen with L-pampo induces robust humoral responses. (**A**) A schematic of the immunization strategy. BALB/c mice (total n = 42; n = 6/antigen-only; n = 8/Alum, AddaVax, AddaS03, or L-pampo (L); n = 4/CpG–Alum) were immunized with RBD-Fc with or without adjuvants i.m. on day 0 and day 21. (**B**) Neutralizing antibody production on day 35. (**C**) ACE2-blocking antibody titers on day 35. (**D**) Total RBD-specific IgG, IgG1, IgG2a, and IgG2b antibodies on day 35. Data shown are median ± IQR. Each dot represents an individual mouse. GMT in the antibody assay indicates geometric mean titers. Data reflect 2 independent experiments (n = 3/antigen-only group; n = 4/Alum, AddaVax, AddaS03, or L-pampo (L) group; n = 2/CpG-Alum group in single experiment). Asterisks indicate statistically significant differences in comparison to the L-pampo (L) group. * *p* < 0.05; ** *p* < 0.01; *** *p* < 0.001; **** *p* < 0.0001; one way ANOVA with the Tukey’s test.

**Figure 4 vaccines-09-00957-f004:**
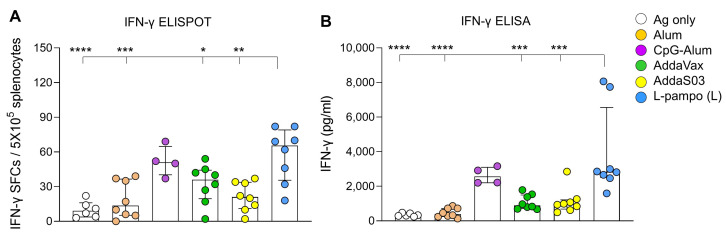
RBD-Fc antigen with L-pampo induces cell-mediated responses. (**A**,**B**) BALB/c mice (total n = 42; n = 6/antigen-only; n = 8/Alum, AddaVax, AddaS03, or L-pampo (L); n = 4/CpG–Alum) were immunized with RBD-Fc with or without adjuvants i.m. on day 0 and day 21 as described in Figure 3A. On day 35, splenocytes were stimulated with the PepMix^TM^ SARS-CoV-2(S-RBD) peptide pool, and RBD-specific IFN-γ secreting cells were analyzed as SFCs by ELISPOT assay (**A**) or total IFN-γ production by ELISA assay (**B**). Data shown are median ± IQR. Each dot represents an individual mouse. Data reflect 2 independent experiments (n = 3/antigen-only group; n = 4/Alum, AddaVax, AddaS03, or L-pampo (L) group; n = 2/CpG–Alum group in single experiment). Asterisks indicate statistically significant differences in comparison to the L-pampo (L) group. * *p* < 0.05; ** *p* < 0.01; *** *p* < 0.001; **** *p* < 0.0001; one way ANOVA with the Tukey’s test.

**Figure 5 vaccines-09-00957-f005:**
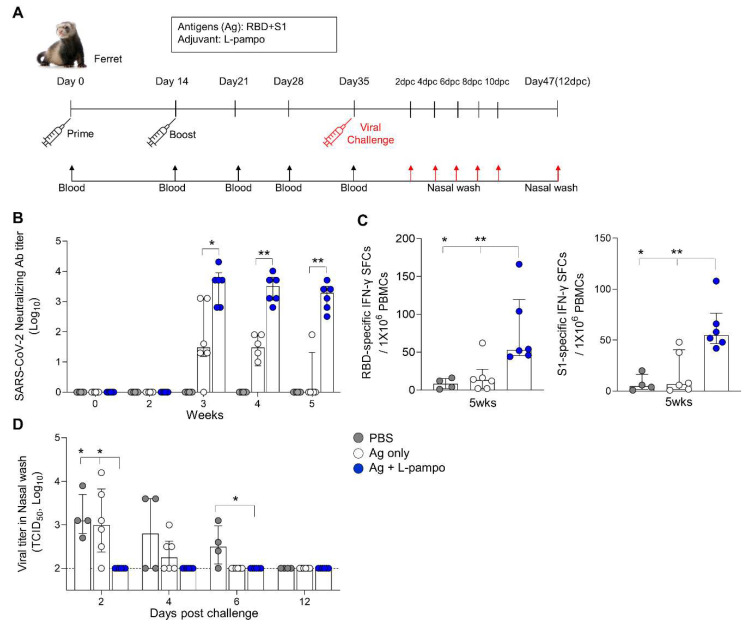
RBD and S1 antigens with L-pampo induce robust humoral and cellular responses in the ferret model. (**A**) A schematic of the immunization and virus challenge strategy in the ferret model. Ferrets (total n = 16, n = 4/PBS; n = 6/Ag only; n = 6/Ag + L-pampo) were immunized with the SARS-CoV-2 antigens (RBD and S1 with 30 μg per each) with L-pampo i.m. on day 0 and day 14. The antigen-only and PBS groups were as controls. On day 35, ferrets were intranasally challenged with 10^5.5^ 50% tissue culture infective doses (TCID_50_/_mL_) SARS-CoV-2 (Korea Centers for Disease Control and Prevention; resource no. 43326) under anesthesia. Blood was collected every week for 5 weeks from the first immunization. After virus challenge, the nasal wash was collected on days 2, 4, 6, 8, 10, and 12. (**B**) Neutralizing antibody production on day 35 prior to challenge of virus. (**C**) On day 35, ferret PBMCs were stimulated with the RBD or S1 antigen and RBD or S1-specific IFN-γ-secreting cells were analyzed as SFCs by ELISPOT assay. (**D**) Viral load in nasal wash on days 2, 4, 6, and 12 after virus challenge by using a TCID_50_ assay. Dotted lines represent lower limits of detection. Data shown are median ± IQR. Each dot represents an individual ferret. Data reflect 2 independent experiments (n = 2/PBS group; n = 3/Ag only or Ag + L-pampo group in single experiment). * *p* < 0.05; ** *p* < 0.01; two tailed Mann–Whiteny test (**B**); one way ANOVA with the Tukey’s test (**C**,**D**).

**Figure 6 vaccines-09-00957-f006:**
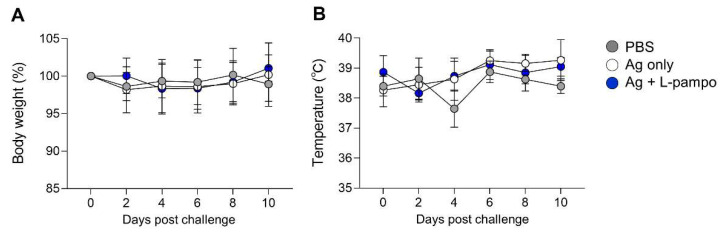
Signs of clinical symptoms after SARS-CoV-2 virus challenge. Immunized ferrets (total n = 16, n = 4/PBS; n = 6/Ag only; n = 6/Ag + L-pampo) were intranasally challenged with 10^5.5^ TCID_50_/_mL_ SARS-CoV-2 (Korea Centers for Disease Control and Prevention, resource no. 43326) under anesthesia. The body weight (%) (**A**) and temperature (°C) (**B**) of ferrets on days 2, 4, 6, 8, and 10 after SARS-CoV-2 virus challenge. Data shown are mean ± SD.

**Table 1 vaccines-09-00957-t001:** Ferrets immunized with L-pampo protect against SARS-CoV-2 virus challenge.

Immunization Group	Virus Replication					
	2 dpc	4 dpc	6 dpc	8 dpc	10 dpc	12 dpc
PBS	4/4 ^1^	4/4	4/4	4/4	3/4	1/4
	(16.8 ± 7.5)	(22.5 ± 2.8)	(21.7 ± 3.9)	(26.8 ± 3.3)	(24.4 ± 3.6)	(27.2 ± 0.0)
Ag only	6/6 ^1^	6/6	6/6	5/6	2/6	0/6
	(21.1 ± 2.2) ^2^	(25.8 ± 3.3)	(24.9 ± 4.5)	(29.6 ± 4.2)	(32.4 ± 3.5)	(0.0 ± 0.0)
Ag + L-pampo	6/6	5/6	3/6	0/6	0/6	0/6
	(31.2 ± 1.9)	(32.7 ± 1.3)	(31.2 ± 0.8)	(0.0 ± 0.0)	(0.0 ± 0.0)	(0.0 ± 0.0)

dpc: day post challenge. ^1^ Number of virus positive mouse/number of total mouse. ^2^ The mean ± SD of cycle threshold (Ct) the value in each group using qRT-PCR.

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
