# Peer review of "COVID-19 Subunit Vaccine with a Combination of TLR1/2 and TLR3 Agonists Induces Robust and Protective Immunity"

_vaccines, 2021, doi:10.3390/vaccines9090957_

Round 1
Reviewer 1 Report
Sometime, the present MS will partly contribute human health in order to control the Covid-19. However, the authors should know the global standards or guidelines about the animal welfare and/or ethics. For example, there are no descriptions about methods for euthanasia of the mice and ferrets used in the protocols in the section "Materials and methods". That is an incredible thing! And adding to this, if they could obtain a registration number issued from an animal experimental committee or something like that set up in among the university, the number should be shown there as well. If there is not such committee, they had better stop to publish their own data including this MS from now.
Author Response
Reviewer 1:
Sometime, the present MS will partly contribute human health in order to control the Covid-19. However, the authors should know the global standards or guidelines about the animal welfare and/or ethics. For example, there are no descriptions about methods for euthanasia of the mice and ferrets used in the protocols in the section "Materials and methods". That is an incredible thing! And adding to this, if they could obtain a registration number issued from an animal experimental committee or something like that set up in among the university, the number should be shown there as well. If there is not such committee, they had better stop to publish their own data including this MS from now.
We appreciate the reviewer’s valuable comment, which we completely agree with. In the revised manuscript, we provided an additional description in terms of the method for euthanasia of mice and ferrets in Materials and Methods as follows.;
Mice were euthanized by the exposure to carbon dioxide (CO2) overdose, followed by cervical dislocation as a secondary method of euthanasia. For ferrets, general anesthesia was induced by injection with 7 mg/kg body weight Zoletil (Virbac, France) and 3 mg/kg body weight Xylazine-HCl (Rompun; Bayer HealthCare, Germany) during the sample collection and inoculation of virus intranasally. Ferrets were euthanized by the administration of T-61 (0.5ml/kg, intravenous route) under general anesthesia when the experiment was completed.
Regarding the registration number from an animal experimental committee, we provided in the section of “Institutional Review Board Statement” in the original manuscript. Briefly, all mice were handled in accordance with the standards approved by the Institutional Animal Care and Use Committee (IACUC, #200081) of CHA University. All ferret care was performed strictly according to the animal care guidelines and experiment protocols approved by the Institutional Animal Care and Use Committee (IACUC) of Konkuk University (permit number: KU20074-1). We followed the guideline of Vaccines by providing the animal welfare statement in “Institutional Review Board Statement” section. In the revised manuscript, we briefly noted this statement in Materials and Methods.
Reviewer 2 Report
This paper aims to study L-pampo as an adjuvant for SARS-CoV-2 vaccines.
The paper suffers of some Major Issues that should discussed before considering it for publication in Vaccines.
MAJOR ISSUES:
- Mice study design: information about sample size (total and single experimental group) is lacking. Please add.
- Sample size: in general, information about the calculation of the sample size and power of the study are lacking. Please add.
- Statistical analysis: data are shown as mean and SEM but a non-parametric test was used for comparison between two groups thus suggesting non normality of distributions. Accordingly, descriptive statistics should be reported as median and IQR.
- Lines 229: authors reported that L-pampo showed increased Ab titers respect to ASO3. However, the difference is not significant (p=0,08) and the sentence is misleading for readers. Please change accordingly.
- In general, the range and distribution of data from L-pampo experiments seems to be quite different respect to compared adjuvants. In particular, SDs/IQRs seem to be quite wide as well as some outliers could be present. Please comment, report detailed values (at least in supplementary tables) and recognize this point as a potential limitation.
- Lines 250-258 and 277-283: these sentences should be removed from Results section and/or moved to the Materials and Methods section.
- Lines 262265: these sentences should be removed from Results section and/or moved to the Discussion section.
- Table 1: Immunization group column should be revised.
Reviewer 3 Report
The pandemic of SARS-CoV-2 has caused over 4 million deaths and over 200 million confirmed cases by now, remaining a rather concerned global health problem. Vaccines with long-term efficacies and sufficient safety are urgently needed. This article attempted to evaluate the immune efficacies of SARS-CoV-2 subunit vaccine adjuvanted with a novel adjuvant consisting of TLR1/2 and TLR3 agonists. The authors found that RBD as well as RBD-Fc antigen, when admixed with L-pampo, could induce potent both humoral and cellular immune response without observed clinical symptoms in both BALB/c mice model and ferret model.
In general, the manuscript demonstrated that L-pampo could serve as a candidate adjuvant in developing new vaccines for SARS-CoV-2. The analysis is appropriate and the conclusions are mostly justified by the data.
However, several questions and suggestions need to be considered:
1. Why did the authors choose S1+RBD as immunogen for BALB/c mice rather than RBD antigen alone, considering the subsequent experiments principally adopted RBD antigen to detect SARS-Cov2 specific IgG and IFN-γ producing T cells?
2. Statistical analysis and authors' evaluation on immune modulating efficacy between L-pampo low and high groups is needed since the authors set these two groups expressly yet subsequently chose L-pampo low to adjuvant with RBD-Fc in Figure 3&4.
3. In table 2, when signs of clinical symptoms were compared in ferret model between experimental and controlled groups, would it be clearer by displaying the data in forms of line charts with appropriate statistical analysis?
4. Besides viral titer in nash water, is there any other indicator that could reflect the protection effects of RBD-S1 adjuvanted with L-pampo, such as histopathology (H&E) of lungs?
